# Cautionary Remarks on the Planetary Boundary Visualisation

Miguel D. Mahecha[1,2,3], Guido Kraemer[1], and Fabio Crameri[4,5]

[1]Institute for Earth System Science and Remote Sensing, Remote Sensing Centre for Earth System Research (RSC4Earth), Leipzig University, 04130 Leipzig, Germany
[2]German Centre for Integrative Biodiversity Research (iDiv), Halle-Jena-Leipzig, Germany
[3]Center for Scalable Data Analytics and Artificial Intelligence (ScaDS.AI), Dresden-Leipzig, Germany
[4]Undertone.design, Bern, Switzerland
[4]International Space Science Institute, Bern, Switzerland

**Correspondence:** Miguel D. Mahecha ({miguel.mahecha, guido.kraemer}@uni-leipzig.de)

**Abstract.** The Planetary Boundary (PB) concept has captured attention across academia and the public alike. Its unique visual representation has been key to the development of the concept and its dissemination. In this commentary, we outline three areas of concern to facilitate further enhancement in the PB concept's visualisation. First, the radial bar plot leads to a quadratic scaling of the effect sizes. Second, the colour gradations denoting the risk of each boundary transgression use complex non-linear patterns, which complicates interpretation. Third, non-linearly distorted colour scales and their fading make the visual perception for people suffering from colour-vision deficiency even more challenging to impossible. The conjunction of quadratic effect scaling and specific colour coding may unintentionally amplify the perception of high-risk areas. We recommend a careful revision of the visual language employed in PB communication. Addressing these concerns will make the PB visualisation a more accurate base for decision makers.

## 1 Introduction

Our planet faces multifaceted pressures, as corroborated by comprehensive reports like those from the IPCC on climate change and the IPBES on biodiversity change (IPBES, 2019; IPCC, 2023), encapsulating numerous additional human-induced Earth system changes. The Planetary Boundary (PB) concept (Rockström et al., 2009a, b) was designed as a framework that provides a unified perspective on the effects of altering various Earth system dimensions. It identifies thresholds, termed 'Planetary Boundaries', within which humans and other organisms can coexist sustainably, thus, ensuring the preservation of Earth's vital life-support systems. With its clarity, the PB concept has emerged as a widely recognized tool for communicating the global change challenges of our era to decision-makers (Steffen et al., 2015). However, given its broad scope, it is not surprising that the PB concept has sparked debate and controversy (Montoya et al., 2018; Rockström et al., 2018; Biermann and Kim, 2020). In response, the PB concept has seen refinements. More recent interpretations address initial omissions of interactions among boundaries (addressed in Steffen et al., 2015) and the absence of spatial mapping (introduced in Richardson et al., 2023). The remaining critiques are summarized by Tandon (2023). However, our aim here is not to critique the PB concept; for that, we redirect readers to the pertinent literature. Instead, we shift our focus to another facet that has thus far remained unaddressed in the discussion: the visual language used to communicate the PB concept.

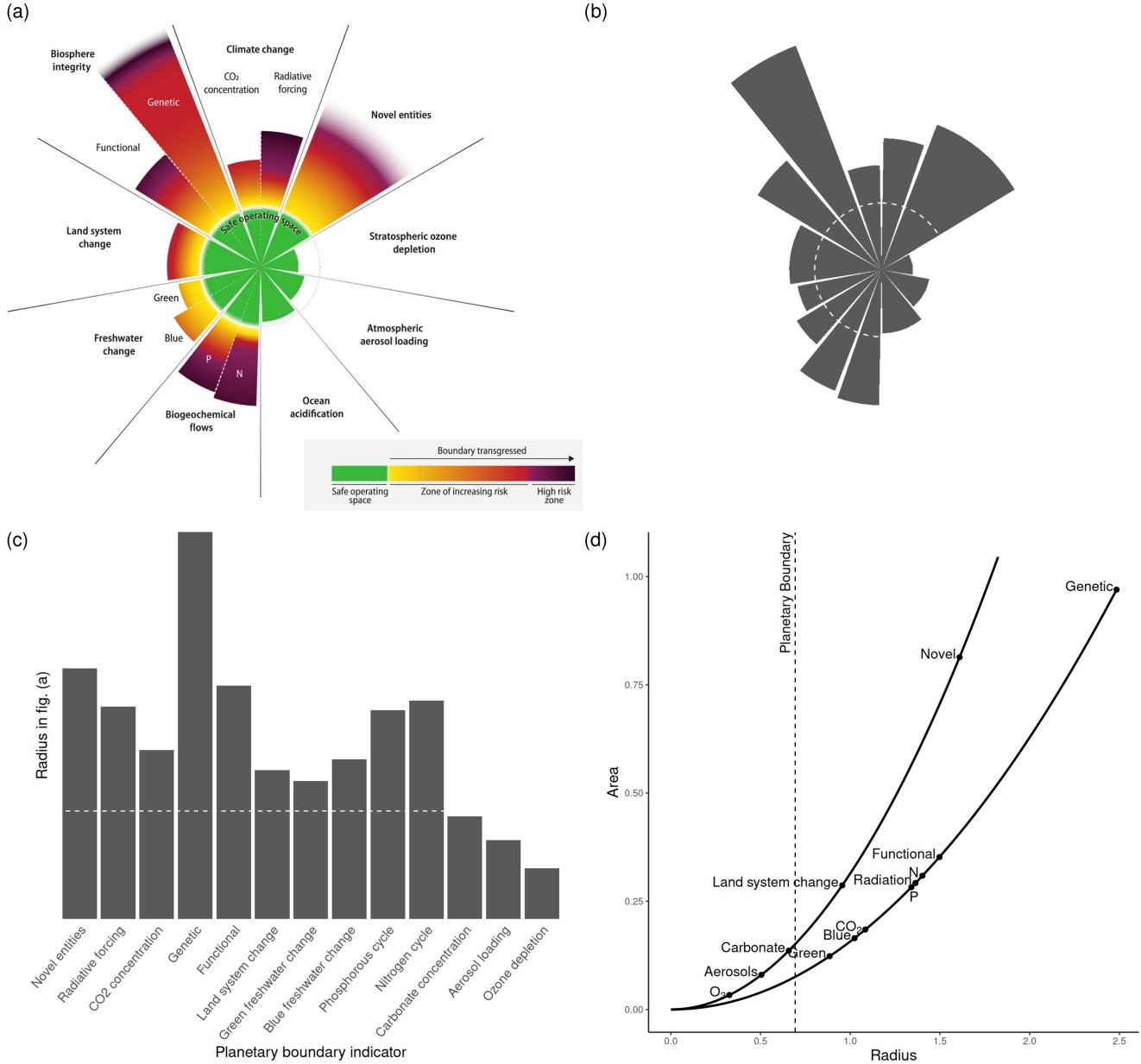

**Figure 1.** Scaling issues inherent in the PB concept visualisation: (a) The latest representation of the planetary boundary concept as a radial bar plot, where indicators representing different boundaries are shown as wedges. The radius of the wedge represents the value of the indicator. Source: Richardson et al. (2023), extracted from the article pdf file, figure released under the Creative Commons Attribution NonCommercial License 4.0 CC BY-NC, https://creativecommons.org/licenses/by-nc/4.0/. (b) The area representation of the PBs as depicted in the original figure in (a). A small difference in the scaling is explained in the text. (c) Actual effect sizes as reported in Richardson et al. (2023). (d) Comparison of the reported PB values to the area shown. The scaling exhibits two distinct curves, as some PBs are divided into two segments. This leads to a halving of the value of the angle $\theta$ and respective area.

Since its inception, the PB concept has consistently featured a powerful visualization. Geere (2020) recounts an intriguing backstory. The conceptual seed for the PB concept, as described by him, was sown by Bo Ekman, founder of the Tällberg Foundation. Ekman envisioned the Earth as a crucial stakeholder at every negotiation table. This figurative idea was then scientifically articulated by Johan Röckström and underpinned by the first PB figure in Rockström et al. (2009b). However, it was likely the version in Rockström et al. (2009a), heavily revised by Wesley Fernandes, an art director with Nature, that made a striking impact, as noted by Geere (2020). This figure employs a radial bar plot, with each wedge representing a different PB and the safe operating space marked by a constant radius. Morseletto (2017) conducted an in-depth analysis of this figure and concluded that it serves as a prime example of science communication, being understandable, meaningful, and engaging. This perception appears to be widely shared, as the figure has been reproduced and adapted extensively (e.g. Nash et al., 2017; Persson et al., 2022; Bachmann et al., 2023), and the concept has even been transferred to other branches of science, such as describing the dimensions of forest disturbance attributes (Turner and Seidl, 2023).

The most recent iteration by Richardson et al. (2023, as reproduced in fig. 1a) introduces a substantial extension. It adds colour gradients intended to illustrate how the transgression of PBs translates into escalating risks. Also this figure has been showcased and replicated by numerous global news outlets, achieving vast reach. At first glance, the figure seems to provide an intuitive visualization of the core messages of the PB concept as interpreted by Morseletto (2017). However, we contend that the current graphical representation of the PB concept and its derivatives, though visually compelling, may inadvertently mislead its audience. Here, we scrutinise the visualisation techniques employed in the PB concept and discuss potential pitfalls and enhancements. Our aim is to initiate a discussion towards developing a visualisation strategy in which the visual language accurately conveys the underlying scientific concepts.

## 2 Scaling of effect size

The figure in question (fig. 1a) presents a radial alignment. In fig. 1b, we recreate the latest figure presented in Richardson et al. (2023), by omitting the risk indicators that are depicted as colours. This variant focuses on the effect sizes, a uni-dimensional value—the distance to the centre. Tabular data of this kind would typically be represented as a bar chart, as illustrated in Fig. 1c. However, due to the radial configuration, the displayed area scales quadratically with the intended value of the variable:

$$A_r = 0.5r^2\theta \propto r^2 \tag{1}$$

where $A_r$ is the area of the wedge, $r$ is the radius (i.e., the value of the PB indicator variable), and $\theta$ is the angle of the wedge, as shown in Fig. 1c. The area of the wedge is perceived as visual weight, which can cause the visual impression conveyed by this plot to not accurately reflect the underlying data, a distortion effect well-documented in the scientific visualisation literature (see e.g. Spence and Krizel, 1994). In chapter 2 of his seminal work, 'The Visual Display of Quantitative Information', Tufte (2001) addresses the general problem of visualisations where the size of the effect scales differently in the visualisation compared to the data, advocating for representations where the size of the effect shown in graphics is proportional to the size of the effect in the data.

That the scaling is a particular issue for radial bar plots, known by various names such as 'radial bar chart', 'radar graph', 'nightingale glyph', 'rose diagram', and 'polar-area diagram', is actually known since their inception by Florence Nightingale (Nightingale, 1858). Distinguished as a pioneer in statistical graphics (among other disciplines), Nightingale depicted deaths in British military hospitals during the Crimean War (1854-56; Cohen, 1984; Brasseur, 2005). Aware of the inherent scaling challenges, she opted for the wedge area rather than the radius to represent the effect sizes of the data (Cohen, 1984), a choice that could indicate the way for an alternative approach to visualizing the PB concept. However, the efficacy of radial charts is debatable. Waldner et al. (2019), for instance, shows that radial charts may be less intuitive for human interpretation compared to Cartesian coordinate systems, even for naturally cyclic patterns such as diurnal or seasonal events.

## 3 Scaling of colour map

The PB figure (Fig. 1a) is colour-coded to show the risk associated with a transgressed PB. This method is reminiscent of the different assessment reports by the IPCC, where so-called 'burning embers' visualise the risks from climate change for various aspects/sectors under different global warming scenarios. These 'burning embers' have also generated considerable attention (for a review on their development, see Zommers et al., 2020) by indicating that certain levels of global warming lead to high-risk zones in specific sectors or impact domains. However, the representation of risks by the PB figure is notably more complex, which prompts the question: why is this the case?

The colour map employed is derived from Inferno (van der Walt and Smith, 2020). Inferno is a colour-map that has been widely adopted and is considered an excellent choice for a continuous colour scale due to its visual uniformity (see Fig. 2a), meaning the perceived difference between colours is proportional to the difference in the values they represent (Crameri et al., 2020). In Fig. 2, we have extracted the colour gradients of the risks associated with the transgression of the PBs and displayed them against the value of the PB itself. To quantify this relationship, we have applied the following formula:

$$r = \log_e \left( \frac{x - \overline{x}_{\text{holocene}}}{x_{\text{PB}} - \overline{x}_{\text{holocene}}} + 1 \right), \tag{2}$$

to reconstruct $r$, the radius of the corresponding wedge in the PB visualisation by Richardson et al. (2023) as a scaling factor, where $x$ is the PB indicator variable, $\overline{x}_{\text{holocene}}$ is the Holocene mean of the PB indicator variable, and $x_{\text{PB}}$ is the threshold defined as planetary boundary. This normalization places the planetary boundary at $\log_e(2)$ and the Holocene mean at $0$[1]. Values have been taken from tab. 1 in Richardson et al. (2023)[2]. The $y$-axis in Fig. 2 is the cumulative distance along the colour gradient in CIELAB2000 colour space (Sharma et al., 2005; Sánchez Beeckman, 2021).

Fig. 2 shows that the risk scales for each PB in a very different manner, and the non-linearity of the scaling is not comparable. For instance, the PB 'Biosphere integrity/Genetic' is the most overshot boundary and the one where the high risk zone is furthest

---

[1]As the viewer can see in Fig. 1a and b, the ratio between the end of the wedges, the centre and the planetary boundary does not quite match the one in Fig. 1a. We also set the current value of the 'Genetics' wedge to 110 E/MSY (extinctions per million species-years), Richardson et al. (2023) give $> 100$ E/MSY as a value but their figure seems to depict a value very close to 100 E/MSY, which makes this bar appear a little bit larger in Fig. 1b

[2]We noted that there seems to be a numerical error in the original visualisation: The pre-industrial Holocene values for blue and green water appropriation have to be either 0% (they are 9.4% and 9.8% respectively) or their wedges in the figure have to be much longer.

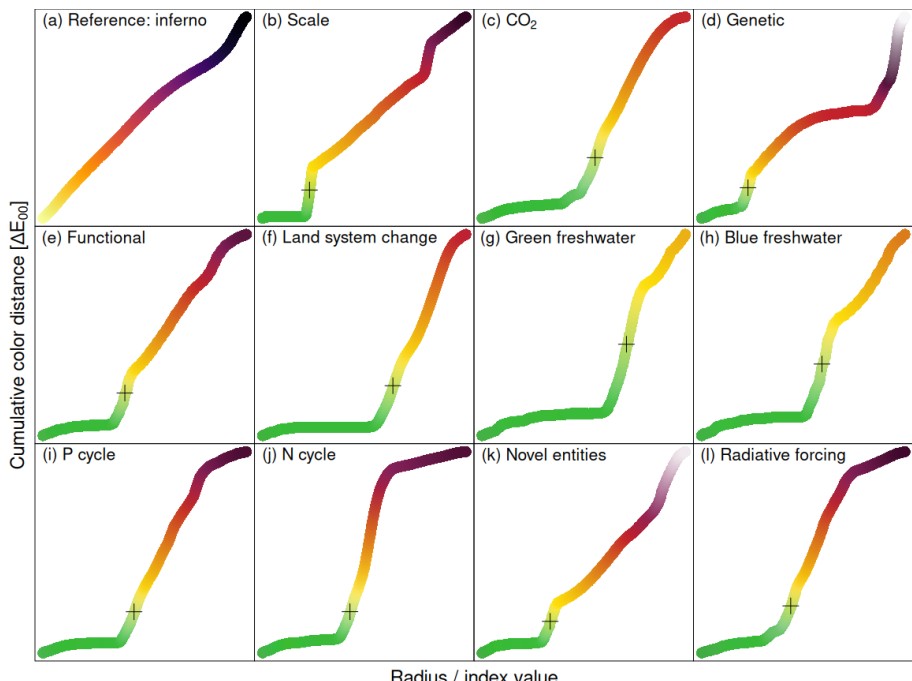

**Figure 2.** The plots show the distance of the indicator along the transgression of the PB vs. the risk as shown as distance in perceptual colourspace (CIEDE2000, Sharma et al. 2005, Sánchez Beeckman 2021). (a) Colour map "Inferno" as reference for a linear colour space, (b) We show the "Scale" of the colour map shown on the bottom of Fig. 1a. In subplots (c) to (l) we show how the cumulative colour distance grows as a function of the change in the PB. Data has been extracted from Fig. 1a.

away but the yellow–red gradient still shows mostly red and very little yellow (compare Fig. 1a and Fig. 2d). In case of the PB
'Biogeochemcial flows/N', the gradient is the one furthest in the purple high risk zone, but the yellow–red gradient is mostly yellow (compare Fig. 1a and Fig. 2j). Moreover, the transition from red to purple on the colour scale is much more abrupt than anywhere else in the circular bar plot. Such variations in scaling are not clarified by Richardson et al. (2023). While we assume that the authors have quantified these transitions, we suggest that this form of visualising them is too subtle and does not allow the viewer to properly quantify the risk progression from the plot.

Another point that adds to the confusion is the seemingly arbitrary order of bars, in the 'burning embers' diagrams, the bars are arranged according to the intensity of the associated risk, which provides a clear gradient of risk, thus facilitating more straightforward visual interpretation. Conversely, the PB visualisation lacks this arrangement and uses a highly non-linear and less transparent scaling of risk, potentially complicating the visual interpretation.

**Colour-vision deficiency (CVD) simulations**

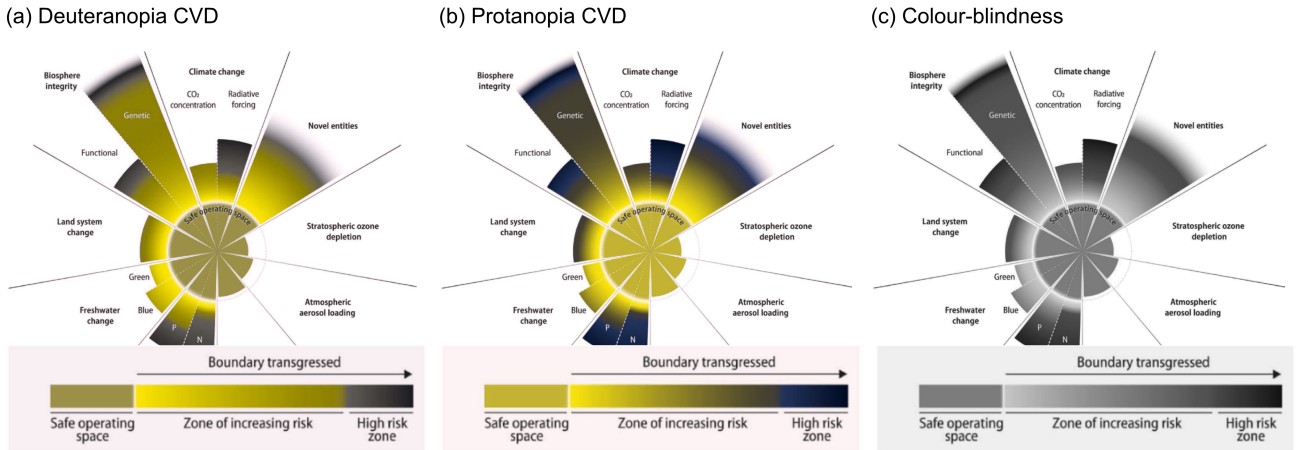

**Figure 3.** Colour-vision deficiency simulations for (a) deuteranopia (green-blindness), (b) protanopia (red-blindness), and (c) full colour-blindness after Brettel et al. (1997) outlining how the currently applied colour coding in the PB figure is inaccessible to some readers.

## 4 Visual accessibility for colour-vision deficiency

Human colour-vision varies among individuals. Most of the population has trichromacy, meaning they possesses three types of cone cells sensitive to long, medium, and short wavelengths of the light spectrum. However, some individuals have fewer functioning types of cone cells, resulting in dichromacy or monochromacy. Trichromatic colour vision is understood well to the extent where perceptual colour spaces, such as CIEDE2000 (Sharma et al., 2005), have been created (as applied, for example, in Fig. 2). Similarly, colour-vision deficiencies (CVD), or absolute colour-blindness, can now be simulated well enough to

detect and prevent accessibility issues (Machado et al., 2009).

Fig. 3 presents simulations of the original PB figure as perceived by individuals with deuteranopia, protanopia, and in grey-scale (representing full colour blindness). These simulations are based on the algorithm from Brettel et al. (1997) and use code from Kovesi (2017). As is well known by graphic design and data visualisation experts, green and red hues often present readability challenges for those with CVD. Using both colours in one figure should be avoided. The colour choice in the current

PB figure design is no exception. Neither individuals with deuteranopia nor those with protanopia can clearly distinguished the "Safe operating space" from the "Zone of increasing Risk" based on their colour alone.

This analysis shows that the current planetary boundaries (PB) figure, characterised by its uneven colour gradients and fading at the edges, fails to provide equal accessibility for individuals with colour vision deficiency. This is an unfortunate oversight for a scientific figure intended to inform policy-making. The importance of universally accessible colour choices, along with

effortless, ready-to-use solutions, has been previously discussed, for example in Crameri et al. (2020).

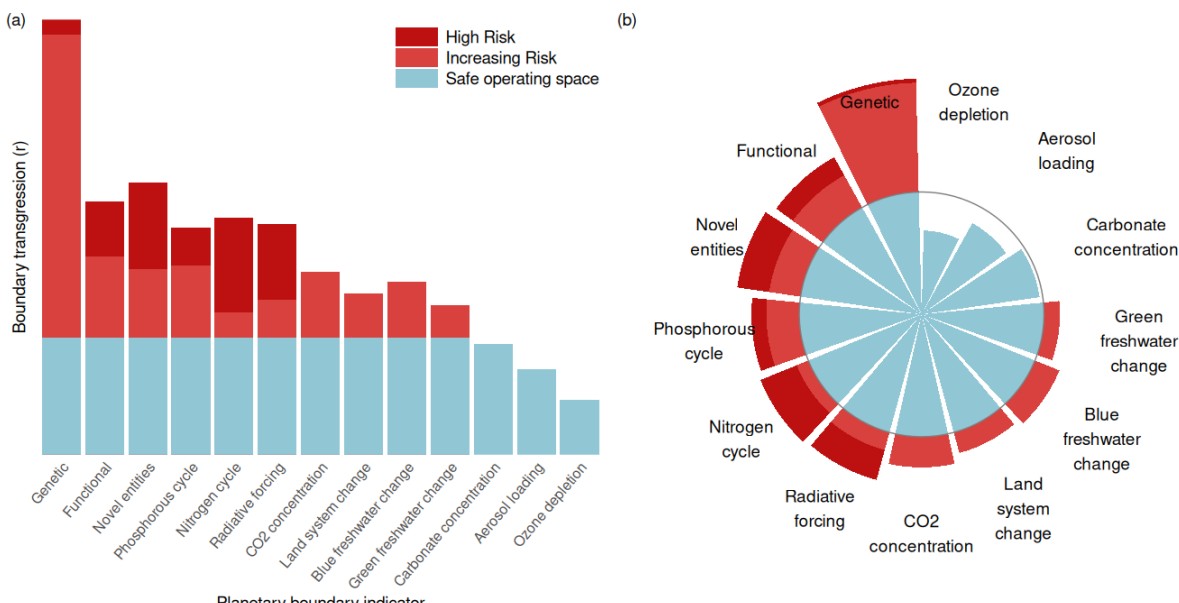

**Figure 4.** Two approaches to alleviate the PB visualisation issues: (a) Translating the PB figure into Cartesian coordinates and choosing a discreet colour bar - visually unattractive, but precise. The bars have been reorganized approximately by size. (b) Maintaining the circular bar chart, but scaling the wedge area by the effect sizes of the underlying variable. Colours are discrete risk levels and the colour scale has been changed so to be accessible to readers with CVD.

## 5    Way forward

Upon analyzing the current PB figure, two logical alternatives for visualising the PB concept emerge and shown in Fig. 4. The first option is choosing a Cartesian coordinate system (Fig. 4a) to avoid issues related to quadratic scaling. Alternatively, if a radial bar plot is preferred, adopting Nightingale's approach of scaling wedge areas should be considered (Fig. 4b) to rectify the
scaling distortion. Additionally, substituting the continuous and complex colour scale for a discrete one yields an unambiguous visual representation. Non-conflicting discrete colours also minimize the misconception risk for people with a dichromacy or monochromacy visual perception. As stop light colour coding might be misconceived (Geere, 2020), we choose here two red colours that show the transgression region and a neutral colour for the within-boundary level.

In both alternatives, we have arranged the bars in a rough order of decreasing transgression while maintaining group cohe-
sion. Our sketch does not depict the variable grouping. While these visualisations may not possess the artistic elegance of the original, they should convey the underlying data more precisely. Of course, Fig. 4 also has limitations, such as the simplified portrayal of increasing risk and the omission of uncertainty visualisation. However, proposing a ready-to-use alternative figure is outside the scope of this commentary. Future versions of the PB concept could, for instance, also consider incorporating interactive features to explore various future scenarios. Today, tools for interactive, web-based data analysis are ubiquitous

and some are capable of handling big data (for spatiotemporal data see e.g. Söchting et al., 2023). But there also many other avenues are thinkable. Our intent here is simply to spark a dialogue on the visual representation of the PB concept.

## 6 Conclusions

In analysing the radial visualisation of the Planetary Boundary concept in the version published by Richardson et al. (2023), we note three areas of concern. Firstly, the quadratic, area-based scaling effect may amplify the perceived transgression of
PBs. Secondly, the highly non-linear risk mapping functions used could potentially complicate the interpretation. Thirdly, the current planetary boundaries figure lacks visual accessibility for individuals with colour vision deficiency. Considering the interplay between area and colour perception (Solso, 1994), an additional issue may emerge: The darker, high-risk colours might compound the quadratic effect—an effect that is, however, very hard to quantify. All these issues undermine the effectiveness of the PB figure in informing policy-making. Given the PB concept's aim to assist decision-makers (Steffen et al., 2015), we
advocate for the development of a more precise visual language. The 'burning ember' approach (Zommers et al., 2020) presents one possible alternative. Exploring other visualisation approaches, such as two-dimensional plots or an ordinal discretization of the colour scale, could also be considered.

*Code availability.* Figures generated by the authors can be reproduced from https://zenodo.org/doi/10.5281/zenodo.10182293, the code is also available under https://github.com/gdkrmr/Cautionary-Remarks-on-the-Planetary-Boundary-Visualization-Supplementary-Materials/

*Author contributions.* M.D.M. and G.K. conceived the paper and wrote the first draft. F.C., initially acting as a journal reviewer of the first draft, joined the writing team due to a significant contributions, i.e. the CVD simulations (Fig. 3) that were incorporated into the paper. All authors revised the final version of the paper

*Competing interests.* No competing interests.

*Acknowledgements.* We extend our gratitude (in alphabetical order) to Sarah Cornell, Jonathan Donges, Wolfgang Lucht and Katherine
Richardson, authors of Richardson et al. (2023), for their constructive feedback on an early version of the paper. Their insights and hints to related materials greatly assisted in better contextualizing our paper. We also appreciate valuable comments from Ana Bastos, Friedrich Bohn, Gustau Camps-Valls, Ida Flik, Christian Wirth. We thank reviewer 1 and EIC Axel Kleidon for supporting us in the development of the final paper. We employed artificial intelligence tools to refine the wording and grammar of our manuscript.

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
