# Peer review of "Cautionary Remarks on the Planetary Boundary Visualisation"

_EGUsphere, 2023_

## Referee Comment (RC2)

Colour-vision-deficiency simulation
*Deuteranopia*

[Figure]

Colour-vision-deficiency simulation
*Protanopia*

---

## Author Response (AR1)

**RC1**: , Anonymous Referee #1, 29 Jan 2024

**This was an interesting and informative short of paper on what has become a very influential representation and framing of anthropogenic impacts: the planetary boundaries. The authors successfully establish two areas for potential improvement, namely area and colour scaling. Their discussions are informed by relevant literature and the suggestions they propose address the issues they identify.**

We thank the reviewer for the assessment; great to see that our arguments seemed to resonate.

**One possible critique that could be levelled at the manuscript is that the planetary boundaries concepts is fundamentally qualitative, and so detailed discussions around its visualisation could be moot. However, there are values assigned to different planetary boundaries, and so they should be represented in ways that are consistent with such differences.**

We thank the reviewer for the insights and agree: If PBs were only qualitative, no visualisation based on quantitative model-data results should be made. However, given that much of the recent work on improving the PB concept has focused on quantifying them a visualisation that reflects the quantitative evidence is necessary. And such a figure must then be correctly reflecting the processes under scrutiny.

**It may also be argued that the risks emerging from transgressing planetary boundaries are (fundamentally) non-linear and so the greater increase in area with increasing distance from the centre in some sense communicates that non-linearity. However, again, that would be to go beyond values assigned to the planetary boundaries - in effect to make potentially unwarranted inferences.**

While the risks are certainly non-linear in nature (hence 'boundary'), a distinction could be made between the risk of changes in the behaviour of the current Earth system and the risk imposed on human civilisation by these changes. In any case, our point is that the nonlinearities in the figures are rather arbitrary.

**Given that an important motivation to graphically represent the planetary boundaries, is effective communication of risk, it would be very interesting to see quantitative data collection and analysis that would evaluate people's responses to different representations. This would serve as an interesting alternative but complimentary approach developed in this manuscript. Such possible work is outside the scope of the manuscript, but could perhaps be included in future work or discussion sections.**

This is a great idea. We think it would be an interesting exercise to actually quantify the effect of the visual distortion but that this kind of analysis is not only out of scope for the present manuscript but would also go beyond our expertise and would depend on other scientists.

**My only suggestion is to include text that acknowledges the need to use figure colour charts with colour vision impaired readers.**

We thank the reviewer for the suggestion and are considering extending the manuscript to include an extended discussion vision impairment as it was also suggested by the reviewer #2 who even did a simulation experiment on this aspect. We are in contract with reviewer #2 and editors of whether or not we can include this contribution.

**RC2**: 'Comment on egusphere-2023-2760', Fabio Crameri, 22 Mar 2024

**The manuscript is of high quality, content- and presentation-wise and therefore provides insights and clarity about the methodology behind the timely and important matter of „planetary boundaries". It is well suited for ESD. The authors concisely explain the misuse of visual elements in the representation of how humanity fares compared to these boundaries, and how they are designed in an inaccurate way. The existing graph type and its graphical elements alter the data that is subsequently misperceived by the readers. This should not be the case and is why the authors offer their valuable perspective on the problem and suggest how to fix some of the issues.**

We thank the reviewer for the very positive review and are very happy that we agree on the importance of the issues addressed in the manuscript.

**As another reviewer already pointed out, the authors did not mention that the figure is also inaccessible to readers with some forms of colour-vision deficiency (e.g., deuteranopia – see colour-vision deficient simulations based on Brettel et al.,1997, attached). I wonder whether this was on purpose, or whether the authors want to highlight that particular shortcoming of the figure too? – it certainly would be worth it.**

We thank the reviewer for the suggestion and would suggest extending the manuscript by one section that discusses the perception of colour vision deficient people of the PB visualisation. Thank you very much for going this extra mile and providing simulations of the effect as an attachment.

**Taking a step back here and realising that one of the figure's main intend is to act as a base for policy decision making is concerning: It is both misleading and inaccessible. Even more worrying is the fact that the current figure has been waved through by so many researchers, editors, and science communicators without challenging its design. For academia, this is not something to trumpet.**

We agree with the reviewer that the impact of the PB visualisations, especially on policy-makers makes an accurate representation of the underlying numbers more important. We are also a bit surprised that our critique has not been articulated earlier.

**Like with other misapplications of graphic design tools in science, the original cause of this one also boils down to most academics having never undergone even just a basic**

**education in scientific visualisation, as this topic is commonly lacking from university curricula. As such, the figure stands for a much larger picture that we—as a community—need to tackle sooner or later.**

**The authors here focus solely on the technical faults of the figure, which is fine. It is, however, „only" reacting to the underlying problem, and in truth, preventing the authors from focussing on other aspects of their research.**

**I add my thoughts on this here as I don't know whether the authors want to go there too and discuss the basic, underlying problem. I don't think they absolutely need to (and as a reviewer, I am in no position to tell them to do so) but I wanted to make sure it is mentioned here somewhere. However, if we don't highlight the actual underlying problem, I fear we will have to keep reminding authors of figures (and our peers, reviewers, editors…) of the importance of accurate and accessible figures in the future.**

We agree with the reviewer that this is only one symptom of a wider problem in academia concerning the accurate use of figures and visualisations. However, we are not sure that the scope of this paper is to address the fundamental and general problem. If you want to do that, we think you should take several examples and not present it based on a single study. We wanted to keep the manuscript concise and focus on suggestions to improve this particularly important visualisation.

**Regarding the manuscript at hand, I only have a few more minor points outlined below.**

*SPECIFIC COMMENTS*

**Radial charts are less intuitive for human interpretation compared to Cartesian coordinate systems, but they usually (maybe because they are rarer) attract more attention. Taking a step back from the solely technical side of the graphic, I think the graphic tries to fulfil two key purposes (which I think could be mentioned more prominently in the current version of the manuscript): first, the accurate and fair data representation and, second, the intentional promotion for avoiding overstepping the planetary boundaries. Reaching both goals at the same time is a formidable graphic design challenge (see e.g., Crameri, F., G.E. Shephard, and E.O. Straume (2022, Pre-print), The open collection of geoscience graphics, EarthArXiv, https://doi.org/10.31223/X51P78 ), and neither the current version, nor probably either of the suggested alternatives, has – as the authors also outline – achieved that, yet.**

We thank the reviewer for this insight and agree that achieving accurate data representation as well as making figures have an impact (especially outside of academia) is hard and we will mention this in a future version of the manuscript.

**Line 11: typo - „to provides"**

**Line 48: A reference here would support this statement. Spence and Krizel (1994), for example, nicely outlines how children (and adults) misjudge angles and areas in polar graphs.**

**Spence, I. and Krizel, P. (1994), Children's Perception of Proportion in Graphs. Child Development, 65: 1193-1213. https://doi.org/10.1111/j.1467-8624.1994.tb00812.x**

**Line 81: Comma after „'Biogeochemcial flows/N'" is missing.**

**Figure 2, Caption: „(a) colour map "Inferno" as reference for a linear colour space,,, – I believe the word ‚colour' at the beginning of the sentence should be capitalised and the sentence should end with a full stop instead of a comma.**

We thank the reviewer for the thorough corrections of the manuscript, we will correct the typos and grammatical errors mentioned and add the reference.

**Thanks for an interesting and useful manuscript!**

**Fabio Crameri**

We thank Dr. Crameri the reviewer for this very positive and insightful review and will followed up with him on a personal basis regarding the interesting points raised. In this discussion the reviewer also offered to perform more simulations. We have worked together on these and decided to include them in the paper. Given that with this he has substantially contributed to our work, according to the "Guidelines for Safeguarding Good Research Practice  - Code of Conduct" of the German Research Foundation (DFG) which applies to the first two authors "An author is an individual who has made a genuine, identifiable contribution to the content of a research publication of text, data or software." As this is the case, we included him as coauthor. The contribution is delineated in the respective section.